# Fabricating polyoxometalates-stabilized single-atom site catalysts in confined space with enhanced activity for alkynes diboration

Yiwei Liu [1,2,8], Xi Wu[3,8], Zhi Li[1], Jian Zhang [1], Shu-Xia Liu[4], Shoujie Liu[5], Lin Gu [6], Li Rong Zheng[7], Jia Li [3✉], Dingsheng Wang [1✉] & Yadong Li [1✉]

Effecting the synergistic function of single metal atom sites and their supports is of great importance to achieve high-performance catalysts. Herein, we successfully fabricate polyoxometalates (POMs)-stabilized atomically dispersed platinum sites by employing three-dimensional metal-organic frameworks (MOFs) as the finite spatial skeleton to govern the accessible quantity, spatial dispersion, and mobility of metal precursors around each POM unit. The isolated single platinum atoms ($Pt_1$) are steadily anchored in the square-planar sites on the surface of monodispersed Keggin-type phosphomolybdic acid (PMo) in the cavities of various MOFs, including MIL-101, HKUST-1, and ZIF-67. In contrast, either the absence of POMs or MOFs yielded only platinum nanoparticles. $Pt_1$-PMo@MIL-101 are seven times more active than the corresponding nanoparticles in the diboration of phenylacetylene, which can be attributed to the synergistic effect of the preconcentration of organic reaction substrates by porous MOFs skeleton and the decreased desorption energy of products on isolated Pt atom sites.

[1] Department of Chemistry, Tsinghua University, Beijing, China. [2] Zhang Dayu School of Chemistry, Dalian University of Technology, Dalian, China. [3] Shenzhen Geim Graphene Center and Institute of Materials Research, Tsinghua Shenzhen International Graduate School, Tsinghua University, Shenzhen, China. [4] Key Laboratory of Polyoxometalate Science of the Ministry of Education, College of Chemistry, Northeast Normal University, Changchun, Jilin, China. [5] College of Chemistry and Materials Science, Anhui Normal University, Wuhu, Anhui, China. [6] Institute of Physics, Chinese Academy of Sciences, Beijing, China. [7] Beijing Synchrotron Radiation Facility, Institute of High Energy Physics, Chinese Academy of Sciences, Beijing, China. [8] These authors contributed equally: Yiwei Liu, Xi Wu. ✉email: li.jia@sz.tsinghua.edu.cn; wangdingsheng@mail.tsinghua.edu.cn; ydli@mail.tsinghua.edu.cn

Single-atom site catalysts (SACs), as the size limitation of nano-catalytic materials, have greatly promoted the development of traditional heterogeneous catalysis and synthetic chemistry[1–4]. The peculiar influence of supports on the catalytic behaviors and stability of single metal atom sites are wide studied[5–10]. To acquire stable atomic-scale dispersion, strong metal–support interactions are requisite to suppress the higher surface energy and mobility of single atoms and thereby prevent their aggregation into nanoparticles (NPs)[11,12]. To date, a series of supports have been applied to fabricate SACs, including metal[13], oxides[14–17], graphene[18–21], porous nitrogen-doped carbon[22–27], metal-organic frameworks (MOFs)[28–30], and zeolites[31–33], etc. It should be noted that besides stabilizing SACs and retaining their high dispersion under rigorous reaction conditions, in some cases, supports can also act as regulators or co-catalyst through modulating the electronic state of single metal atoms[34–36]. Hence, developing innovative supports to induce synergistic or stimulative functions with single metal atoms would be an effective approach for fabricating promising SACs.

An ideal support is expected to possess abundant anchoring sites exposed and be functionalized to realize the synergistic function. Based on these considerations, we speculate that polyoxometalates (POMs) would be a qualified candidate. Because, as a kind of well-defined nanoscale metal-oxygen clusters, POMs have several advantages. Structurally, POMs are composed of the corner- and edge-shared metal-oxygen octahedral[37,38], the abundance of surface oxygen atoms provides a range of coordination sites for single metal cation[39,40]. And the charge density and coordination geometry of POMs surfaces are manipulatable by adjusting their composition and structure[41]. As an example, the classical Keggin-type POMs have up to five isomers (Supplementary Fig. 1) that afford various anchor sites for single atoms. In terms of properties, the strong Brønsted acidity and reversible redox behavior of POMs allow them to be used as effective co-catalyst to achieve the synergistic effect with other catalytic active species[42,43]. In addition, the simple synthesis route and monodisperse nanoscale size of POMs provide a platform for further combination with other substrates[44,45]. However, in the synthesis of conventional POM-based metallic nanomaterials, metal precursors and POMs are both simply dispersed in the liquid-phase systems without any space restrictions. Owing to the limitation of the number of coordination sites on POMs surface, when these unsaturated sites are completely occupied by single metal atoms, an excess of metal precursors is prone to agglomerate into NPs until the surface is stabilized by POMs (Fig. 1)[46,47]. Although many efforts have been made, such as reducing the input of metal precursors, the generation of NPs can hardly be completely inhibited due to the spatially inhomogeneous dispersion caused by the high mobility of metal precursors[46,48]. Hence, to avoid their aggregation, in addition to controlling the accessible quantity of metal precursors, it is also necessary to restrict their spatial dispersion and mobility. Specifically, if we could introduce a limited number of metal precursors around each POM unit and guarantee their high dispersity and low mobility, the vagrant single atoms would be immediately captured by POMs when losing the protection of ligands during the reduction process, which is expected to be achieved in confined space.

Based on these principles, we elaborately selected MOFs, featuring three-dimensional periodic spatial matrix structure[49–54], as the spatially restricted domain skeleton to constrain the distribution and migration of platinum acetylacetonate precursors surrounding Keggin-type phosphomolybdate ($H_3PMo_{12}O_{40}$, denoted as phosphomolybdic acid (PMo)). A series of single platinum atomic site catalysts stabilized by PMo were successfully constructed in the various MOFs systems (denoted as Pt$_1$-PMo@MOFs), including MIL-101, HKUST-1, and ZIF-67. In contrast, either the absence of POMs or MOFs yielded only platinum NPs. Remarkably, Pt$_1$-PMo@MIL-101 exhibited catalytic activity far beyond that of Pt NPs in MIL-101 without PMo by a factor of >7 in the diboration reaction of alkynes, which can be attributed to the synergistic effect of the pre-concentration of organic reaction substrates by porous MOFs skeleton and the decreased desorption energy of products on isolated Pt atom sites.

## Results and discussion

**Synthesis and characterization of Pt$_1$-PMo@MIL-101.** The POMs-stabilized single platinum atom sites catalysts were synthesized by reduction of platinum acetylacetonate precursors in spatial confined MOFs skeleton. Typically, Keggin-type PMo was introduced into MIL-101 by an in-situ synthesis method resulting in uniform dispersion of PMo without jeopardizing structural and compositional features of MIL-101 (denoted as PMo@MIL-101)[42,55]. Owing to the size of PMo was bigger than the apertures of the small cages (Cage A) but smaller than that of the big cages (Cage B) of MIL-101 (Fig. 2a), the PMo can only occupy part of the skeleton, and the PMo content is ~16 wt% determined by ICP-OES. Comparing with the parent MIL-101, a noticeable change in powder X-ray diffraction (PXRD) patterns of PMo@MIL-101 ~$2\theta = 7°$ was observed (Fig. 2b). This change can be attributed to the ordered PMo assemblies residing in cage A which is in line with previous studies on POMs/MOFs composite materials[55]. Fourier-transformed infrared (FTIR) spectrum (Fig. 2c) complimentary proved the successful injection of PMo into the skeleton, and the characteristic absorption peaks appeared between 800 and 1000 cm$^{-1}$ indicated the intact Keggin structure was not changed during synthesis. The microporous feature was preserved in PMo@MIL-101 with a relatively high Brunauer-Emmett-Teller (BET) surface area of ~1419 m$^2$/g (Supplementary Fig. 2), which was beneficial for the spatial dispersion of metal precursors in the skeleton. Platinum acetylacetonate was selected as the precursor owing to its relatively matched molecular size (9.8 Å) with the cages of MIL-101 and PMo. The Pt precursors were implanted into PMo@MIL-101 through a liquid-phase impregnation process. The platinum-containing PMo@MIL-101 were then collected and reduced in H$_2$ atmosphere at 150°C to obtain isolated single Pt atom sites on PMo@MIL-101 (denoted as Pt$_1$-PMo@MIL-101).

Transmission electron microscopy (TEM) was used to preliminary investigate the size and morphology of Pt species in the MOF skeleton. As shown in Fig. 3a, Pt$_1$-PMo@MIL-101 retained the octahedral morphology and energy-dispersive X-ray spectroscopy images revealed that Cr, Mo, and Pt elements were homogeneously distributed over the entire MOFs crystals with no NPs observed (Fig. 3b and Supplementary Fig. 3). The Pt content was ~0.24 wt%, determined by ICP-OES. To further confirm the atomically dispersed nature of platinum, Pt$_1$-PMo@MIL-101 was characterized by aberration-corrected high-angle annular dark-field scanning transmission electron microscopy (AC HAADF-STEM). We observed some brighter dots that can be identified as Pt single atoms (Fig. 3c, highlighted by red circles). Extended X-ray absorption fine structure (EXAFS) further provided critical evidence of platinum's atomic dispersion and its interaction with PMo. As shown in Fig. 3e, only one notable peak contributed by PtO-scattering paths was examined in the Fourier-transformed (FT) $k^3$-weighted EXAFS spectrum. The Pt-Pt peak of Pt foil standard at ~2.6 Å was not detected in Pt$_1$-PMo@MIL-101, demonstrating the mere existence of isolated Pt single atoms. The X-ray absorption near-edge structure (XANES) curve (Fig. 3d) of Pt$_1$-PMo@MIL-101 showed a peak height between those of Pt and PtO$_2$, suggesting the Pt atoms possess the positive charge Pt$^{\delta+}$

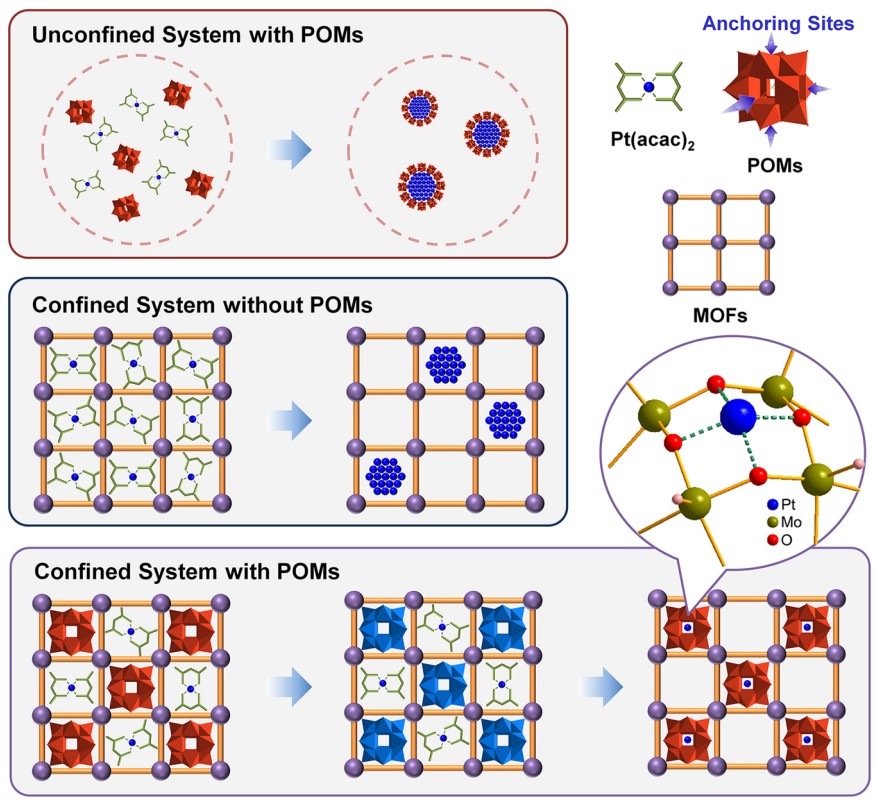

**Fig. 1 Schematic illustration of the fabrication of different Pt nanomaterials.** Pt nanoparticles are formed in the absence of either POMs or MOFs, and POM-stabilized single Pt atom sites can be constructed in the restricted domain space of MOFs.

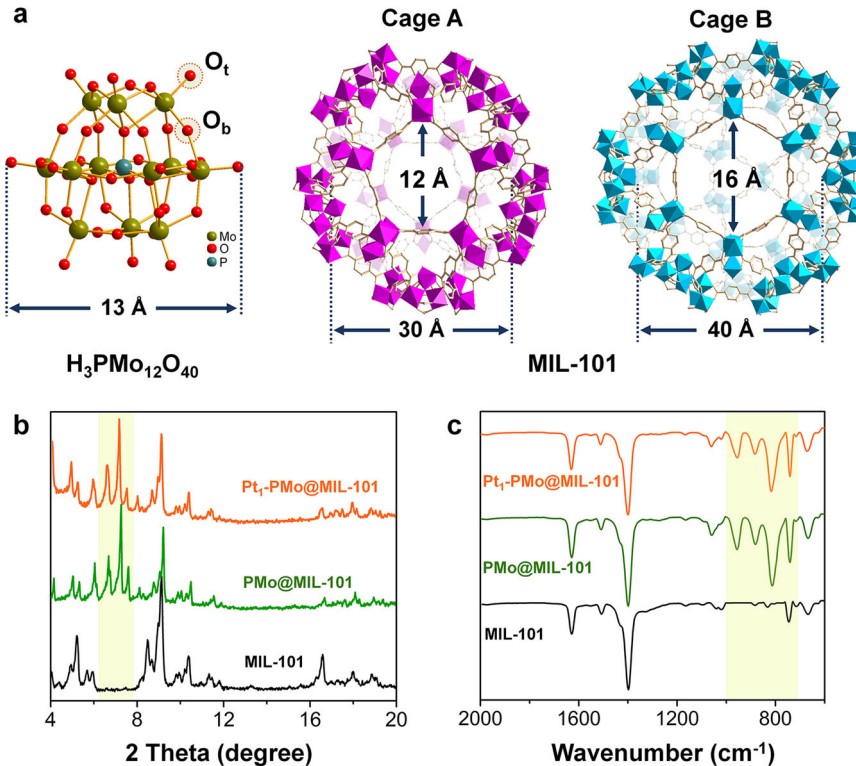

**Fig. 2 Structure and characterizations of $H_3PMo_{12}O_{40}$, MIL-101, and PMo@MIL-101. a** The crystalline structures of Keggin-type $H_3PMo_{12}O_{40}$ and the pore structures and aperture size of MIL-101. $O_t$: terminal oxygen, $O_b$: bridge oxygen. **b** PXRD of MIL-101, PMo@MIL-101, and $Pt_1$-PMo@MIL-101, respectively. **c** FTIR of MIL-101, PMo@MIL-101, and $Pt_1$-PMo@MIL-101, respectively.

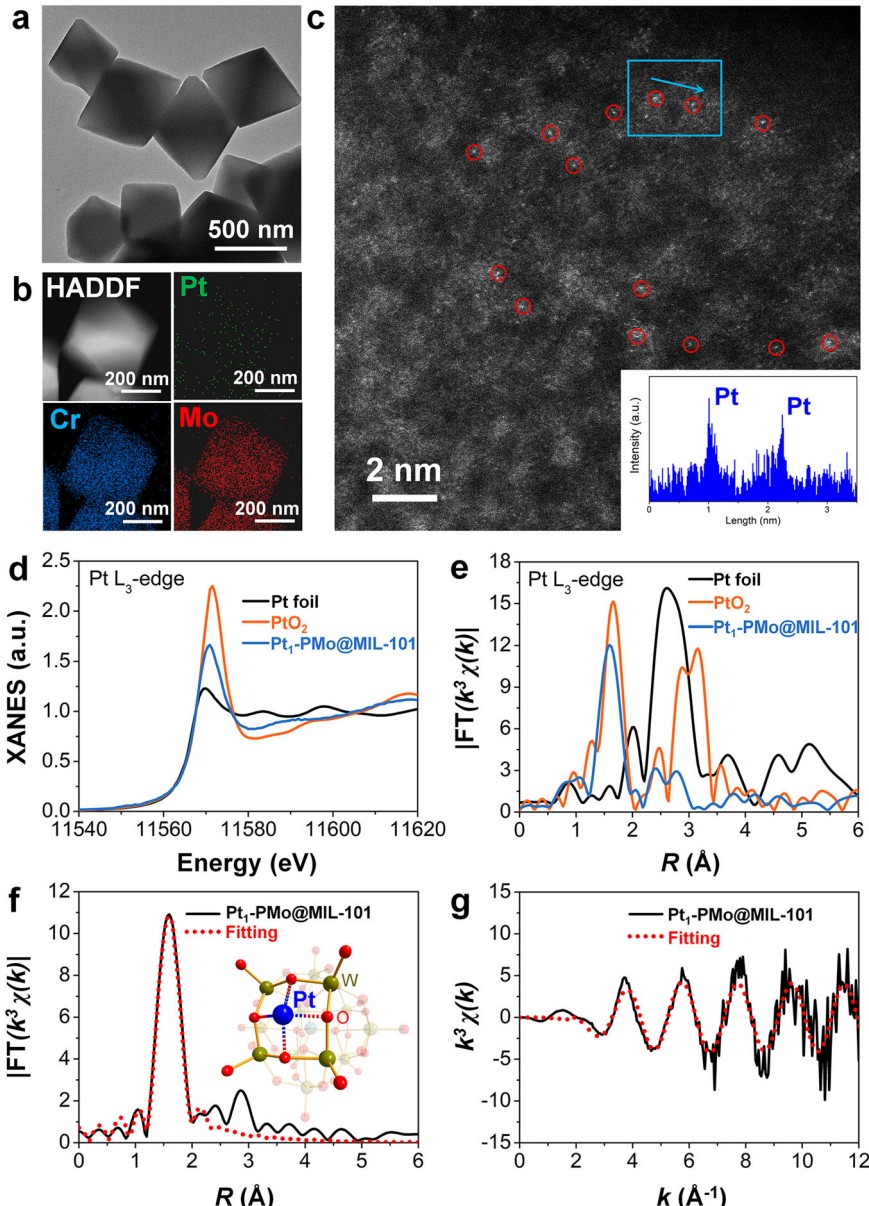

**Fig. 3 Characterizations of Pt₁-PMo@MIL-101. a** TEM images. **b** HAADF-STEM images and corresponding element maps showing distributions of Pt (green), Cr (blue), and Mo (red), respectively. **c** Aberration-corrected HAADF-STEM images and corresponding Z-contrast analysis. **d** XANES of Pt $L_3$-edge for Pt₁-PMo@MIL-101 with reference materials Pt foil and PtO₂. **e** Fourier transforms of $k^3$-weighted Pt $L_3$-edge EXAFS experimental data for Pt foil, Pt₁-PMo@MIL-101, and PtO₂. **f** EXAFS-fitting curves of Pt₁-PMo@MIL-101 at R space and the inset is the proposed model for Pt₁-PMo@MIL-101. **g** EXAFS-fitting curves of Pt₁-PMo@MIL-101 at k space.

rather than Pt⁰. Furthermore, EXAFS fitting was carried out to obtain the quantitative structural configuration of Pt in Pt₁-PMo@MIL-101. A square planar coordination geometry structure constructed by four bridging oxygen on the surface of PMo was proposed as the most stable anchoring site for a single Pt atom (Fig. 3f, Supplementary Table 1), and this site was also further confirmed through density functional theory (DFT) calculation by comparing the formation energy of the anchored Pt atom (Supplementary Table 2). This configuration was consistent with atomically dispersed platinum on oxides and satisfied the empirical requirements of coordination chemistry. The quasi planar coordination geometry of a single Pt atom is key to its catalytic activity. The d orbitals of Pt atom in a uniform square planar field would be split into the $d_{xz}$ ($d_{yz}$), $d_{z^2}$, $d_{xy}$, and $d_{x^2-y^2}$ orbitals, indicating that six electrons would occupy the three

lowest energy orbitals with the valence of +4. However, the single Pt atom with four nearest-neighbor O atoms in PMo@MIL-101 has a non-planar configuration. As seen in the cross-section, the distance of the Pt-O bond (2.013 Å) in the y direction is larger than that in the x direction (1.850 Å). This distorted square planar field would lead to an increase in the energy level of $d_{xz}$ and $d_{yz}$ orbitals, and the occupancy of $d_{xz}$ orbital is changed from full-occupied to half-occupied (Supplementary Fig. 4). However, the half-occupied $d_{xz}$ orbital is more active and will benefit the catalytic activity of Pt₁-PMo@MIL-101, which provides the active sites for reactants adsorbed in xz direction (such as bis(pinacolato) diboron molecule).

To clarify the necessity of both PMo and MOFs in the fabrication of isolated single Pt sites, we synthesized Pt samples without PMo or MIL-101 under the same conditions for

comparison. Despite there are six theoretically anchoring sites on each PMo unit, superfluous PMo was added (the molar ratio of PMo/Pt å 10) to investigate whether spatial confinement is indispensable. As we predicted, massive Pt NPs were formed. Element line scan revealed that the surface of Pt NPs was protected by PMo and the average particle size of Pt NPs was ~2–3 nm (Supplementary Fig. 5). Likewise, without PMo in MIL-101, both nanoclusters and NPs formed in the cages and on the outside surface of MIL-101 (Supplementary Fig. 6), which can be attributed to the high mobility of Pt atoms and the shortage of anchoring sites on MOFs skeleton. X-ray photo-electric spectrometry (XPS) was used to explore the interaction between Pt and PMo, and trace the atom trapping process. The binding energy of Pt $4f$ in $Pt_1$-PMo@MIL-101 was higher than that in Pt NPs@MIL-101 (Supplementary Fig. 7) demonstrating that the isolated Pt atom was positively charged, which was consistent with the XANES results. This was reasonable, because when Pt agglomerated into NPs in the absence of PMo its valence was near zero. Hence, Mo $3d$ signals were used for further analysis. The binding energy of Mo $3d$ in $Pt_1$-PMo@MIL-101 was slightly decreased compared with PMo@MIL-101 (Supplementary Fig. 7) indicating the existence of electronic interaction between PMo and the single Pt atom. Considering PMo could be reduced by $H_2$ and the resulted heteropoly blue (reduced PMo, denoted as $PMo_{red}$) can act as both reductant and stabilizer to synthesize metal NPs[46], we proposed a POM assistant reducing mechanism to account for the formation of $Pt_1$-PMo@MIL-101 (Fig. 1). In a typical synthesis process, PMo was firstly reduced in the $H_2$ atmosphere to form $PMo_{red}$. Comparing with the initial PMo, the surface charge density of $PMo_{red}$ was increased which promoted the attraction of Pt precursors around the surface oxygen of polyoxoanion. Then, the attracted Pt precursors were in-situ reduced by $PMo_{red}$ and trapped by the surface oxygen to form Pt-ISAS, the electron interaction between Pt-ISAS and electronegative PMo was proved by the slight shift of Mo 3d XPS spectrum which was beneficial for stabilizing isolated single Pt atoms. During this step, $PMo_{red}$ were oxidized to PMo to form a closed circle. In contrast, without PMo these mobile Pt precursors were directly reduced by $H_2$ and grew together until resisted by the pores of MOFs. We carried out $H_2$ temperature-programmed reduction for $Pt(acac)_2$@MIL-101 and $Pt(acac)_2$-PMo@MIL-101 (Supplementary Fig. 8). The reducing temperature decrease ~30° C after the introduction of PMo, which is consistent with our speculation.

**Catalytic performance evaluation of $Pt_1$-PMo@MIL-101**. In order to evaluate the catalytic performance of $Pt_1$-PMo@MIL-101, we chose the diboration reaction of alkynes[56,57] as a model reaction based on the consideration of the inherent catalysis of Pt and the unique component and structural features of MOFs. Comparing with the few reported Pt-based heterogeneous catalysts for diboration reactions whose supports are usually oxides or hydroxides with low catalytic efficiency (turnover frequency < 50 $h^{-1}$)[58,59], the pores of MOFs would act as nanosized reactors to accelerate the reaction via pre-concentration effect owing to the affinity between the organic substrates and the ligands of MOF skeleton[60–62]. As we expected, the diboration reaction of phenylacetylene with bis(pinacolato)diboron ($B_2pin_2$) proceeds rapidly when using $Pt_1$-PMo@MIL-101 as the catalyst. Approximately 96% of phenylacetylene was converted in ~2 h, and almost complete conversion was observed after an additional 30 min. Besides the aimed diboration product, bare of other by-products, such as hydroborylated product, were observed by gas chromatography-mass spectrometer (GC-MS) analysis, and the total yield is as high as 98%. We calculated the turnover frequency

(TOF) values based on the platinum content of $Pt_1$-PMo@MIL-101 which reached a high value of 1017 $h^{-1}$. Comparative studies using Pt NPs loaded in MIL-101 (denoted as Pt NPs@MIL-101) and the parent supports MIL-101 and PMo@MIL-101 were performed. The conversion percentage after 3 h under otherwise identical conditions reached only approximately 13% for Pt NPs@MIL-101, and no products were detected for MIL-101 and PMo@MIL-101 (Fig. 4a). Clearly, the active catalytic species was platinum, and the isolated single Pt atoms were much more efficient than NPs in catalysis progress. To shed light on the origin of single Pt atomic site's significant performance improvement over Pt NPs, we performed DFT calculations to understand the reaction mechanism. The energy evolution of $Pt_1$-PMo@MIL-101 and Pt NPs@MIL-101 catalyzed phenylacetylene diboration reactions was investigated by dividing into three steps[63]. To balance the computational cost and accuracy, we simplified the computational model of Pt NPs@MIL-101 to the Pt (111) surface. As shown in Fig. 4c, the first step consists of the adsorption and dissociation of B-B bond in $B_2pin_2$, with the energy decrease of 1.86 eV on isolated Pt atom and 3.65 eV on Pt (111). In the following second step, the phenylacetylene was adsorbed on Pt sites and the dissociated two Bpin were sequentially inserted on acetylenic bond with a further energy reduction of 3.10 eV for isolated Pt atom and 3.17 eV for Pt NPs, respectively. These two steps were both energetically favorable on different Pt species, although a negligible increase in energy was observed during B-B bond dissociation on the single Pt atom (0.10 eV) and the first Bpin insertion on Pt(111) (0.17 eV). However, the desorption of the resulting product in the third step was an endothermic process. Owing to the multiple adsorption interactions between the product and Pt NPs, such as benzene ring and ethylenic bond strongly bonded with a terrace of Pt NPs, the desorption energy was as high as 4.40 eV compared with the corresponding 2.54 eV on the single Pt atom which well explained the better catalytic performance of $Pt_1$-PMo@MIL-101. The stability of the $Pt_1$-PMo@MIL-101 catalyst was also investigated. Both the structural integrity (Supplementary Fig. 9), isolated single platinum atomic sites dispersion (Supplementary Fig. 10), and the activity (Fig. 4b and Supplementary Fig. 11) of the recovered catalyst were maintained after reused over five cycles. No leaching was detected owing to the rigorous imprisonment of PMo in the MIL-101 skeleton.

The remarkable performance of $Pt_1$-PMo@MIL-101 raised yet another issue, that was, whether MIL-101 subsidiarily promote the catalysis progress although there was no activity itself. Hence, hydrophilic inorganic support, Y zeolite, was used to compare with MIL-101. Pt NPs and the single Pt atoms are loaded in the pores of Y zeolite (denoted as Pt NPs@Y and $Pt_1$@Y) through reported methods[33,64], and characterized by PXRD and TEM (Supplementary Fig. 12–14). Both two catalysts presented almost undetectable activities for diboration of phenylacetylene, which was obviously lower than that of Pt NPs@MIL-101 and $Pt_1$-PMo@MIL-101 (Fig. 4a). These results demonstrated that MIL-101 indeed contributed to the catalytic process, although it was catalytic inert for this reaction. Considering that the Pt catalytic activity center was located inside the porous skeleton, and the reaction substrates need to enter the skeleton to contact with Pt sites, to rationalize these observations, we investigated the adsorption abilities of MOF skeletons for phenylacetylene and $B_2pin_2$ under reaction conditions, and the concentrations of reactants on these selected substrates were all below the initial reaction concentrations to simulate the changes during the reaction process. As shown in Fig. 4d, e, MIL-101 exhibited conspicuous pre-concentration ability for both reactants, and this process was monitored by gas chromatography. After adsorption equilibrium reached, the calculated concentrations of reactants

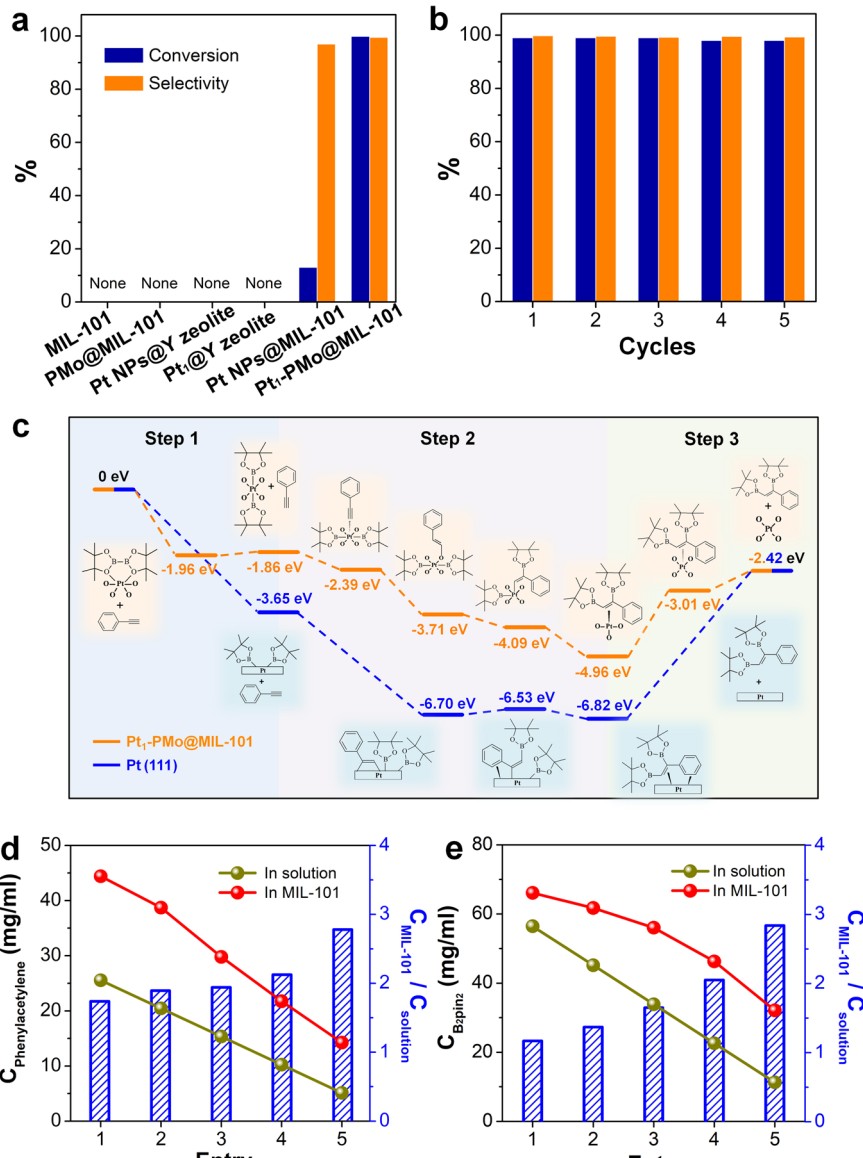

**Fig. 4 Catalytic performance, mechanism, and pre-concentration effect. a** The catalytic activity of phenylacetylene diboration with different catalysts. **b** The cyclic catalytic performance of Pt$_1$-PMo@MIL-101. Reaction conditions: phenylacetylene (0.5 mmol) and B$_2$pin$_2$ (0.5 mmol), catalyst (20 mg), toluene (2 mL) as solvent, $T = 100°C$. Conversion is determined by GC analysis with dodecane as the internal standard. Selectivity is determined by GC-MS analysis. **c** The calculated energy evolution of Pt$_1$-PMo@MIL-101 and Pt (111) catalyzed diboration reactions. **d**, **e** The pre-concentration effect of MIL-101 for phenylacetylene and B$_2$pin$_2$. The concentration and corresponding concentration ratio of phenylacetylene (**d**) and B$_2$pin$_2$ (**e**) in MIL-101 and reaction solutions.

inside MIL-101 were higher than that in the residuary solution, even when the concentration decreased to only one-fifth of the initial. Obviously, in the inner space of the 3D skeleton, phenylacetylene and B$_2$pin$_2$ were "forced" to interact with single Pt atom sites, as they were confined in close proximity. In contrast, the pure inorganic support Y zeolite exhibited negligible adsorption for these two reactants, which should be responsible for the inert catalysis activities of Pt NPs@Y and Pt$_1$@Y. It is possible to attribute the signally disparate adsorption behaviors of MIL-101 and Y zeolite to their different chemical compositions. To be more specific, the organic component (terephthalic acid) of MIL-101 frameworks afforded hydrophobic-hydrophobic and aromatic-aromatic interactions with the organic reactants, which did not exist in Y zeolite. Similar adsorption behaviors were also observed in liquid-phase adsorption desulfurization systems[61].

**Expansion of synthetic system.** Successfully obtained PMo-stabilized isolated single Pt atoms in MIL-101, we intended to expand this strategy to other popular MOF systems with the appropriate porous structure to accommodate Keggin-type POMs as anchoring sites, such as HKUST-1 and ZIF-67 (Supplementary Fig. 15)[65,66]. PMo and platinum acetylacetonate were imported and constrained in these MOFs skeletons with similar synthetic procedures. After the reduction in H$_2$, Pt$_1$-PMo@HKUST-1 and Pt$_1$-PMo@ZIF-67 were successfully synthesized. PXRD (Supplementary Fig. 16 and 17), FTIR (Supplementary Fig. 18 and 19) and N2 adsorption (Supplementary Fig. 20 and 21) proved the structural integrity and the presence of PMo in MOFs skeletons. HAADF-STEM images and elemental mappings of Pt$_1$-PMo@HKUST-1 and Pt$_1$-PMo@ZIF-67 shown in Fig. 5 indicated the homogeneous dispersion of Pt elements. The Pt content was

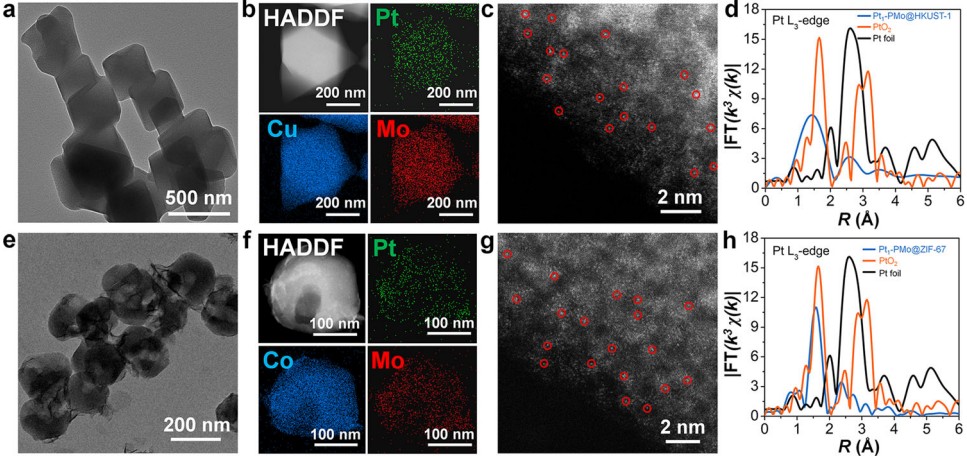

**Fig. 5 Characterizations of Pt$_1$-PMo@HKUST-1 and Pt$_1$-PMo@ZIF-67. a** TEM images of Pt$_1$-PMo@HKUST-1. **b** HAADF-STEM images and corresponding element maps showing distributions of Pt (green), Cu (blue), and Mo (red) in Pt$_1$-PMo@HKUST-1, respectively. **c** Aberration-corrected HAADF-STEM images of Pt$_1$-PMo@HKUST-1. **d** Fourier transforms of $k^3$-weighted Pt $L_3$-edge EXAFS experimental data for Pt$_1$-PMo@HKUST-1. **e** TEM images of Pt$_1$-PMo@ZIF-67. **f** HAADF-STEM images and corresponding element maps showing distributions of Pt (green), Co (blue), and Mo (red) in Pt$_1$-PMo@ZIF-67, respectively. **g** Aberration-corrected HAADF-STEM images of Pt$_1$-PMo@ZIF-67. **h** Fourier transforms of $k^3$-weighted Pt $L_3$-edge EXAFS experimental data for Pt$_1$-PMo@ZIF-67.

determined ~0.41 wt% in Pt$_1$-PMo@HKUST-1 and 0.13 wt% in Pt$_1$-PMo@ZIF-67 by ICP-OES. The AC HAADF-STEM (Fig. 5c–g) and EXAFS (Fig. 5d–h) results indicated that in all cases, the Pt existed in the form of a single atom with positive charges, and the isolated Pt atoms were anchored on the surface of PMo by coordinating with four bridging O atoms. These results forcefully demonstrated the generalization of this spatial confinement strategy for fabricating POMs-stabilized single Pt sites in different MOFs.

In summary, we developed a novel strategy for fabricating POMs-based SACs in the domain-limited space of MOFs. This synthesis strategy is useful in various MOFs systems. Significantly, in the synergy of POM-stabilized single Pt atom sites and MOFs skeleton, Pt$_1$-PMo@MIL-101 exhibited extremely outstanding catalytic performance over conventional Pt NPs in the diboration reaction of phenylacetylene. In consideration of the structural and functional diversity of POMs and porous materials, this strategy has the potential to be extended to other POM-based porous systems, not only limited to MOFs but also to other deliberately selected porous backbone materials, thereby constructing more innovative POMs-based single-atom sites catalysts for purposefully synergistic catalysis.

## Methods

**Synthesis of H$_3$PMo$_{12}$O$_{40}$@MIL-101 (PMo@MIL-101).** The incorporation of H$_3$PMo$_{12}$O$_{40}$ into MIL-101 was realized by adding H$_3$PMo$_{12}$O$_{40}$ during the crystallization process of MIL-101. Typically, 1.0 g Cr(NO$_3$)$_3$·9H$_2$O, 0.42 g terephthalic acid, and 1.0 g H$_3$PMo$_{12}$O$_{40}$ were mixed in 10 mL deionized water and then stirred intensely for 4 h. The pH value of the mixture was adjusted to three by dropwise adding 1 M aqueous NaOH during the stirring process. Then, the mixture was transferred into a Teflon-lined autoclave and kept in an oven at 180 °C for 20 h without stirring. The resulting solid product was washed with DMF three times and then soaked in distilled water for 6 h under stirring. This process was repeated three times to ensure that H$_3$PMo$_{12}$O$_{40}$ was completely removed from the big cages (Cage B) of MIL-101. The resulted product was dried at 80 °C in a vacuum oven for 24 h.

**Synthesis of Pt$_1$-PMo@MIL-101.** In all, 20 mg Pt(acac)$_2$ was dissolved in 10 mL methanol and then 1.0 g PMo@MIL-101 was ultrasonically dispersed in the solution. After continuous stirring for 12 h, the solid was collected through centrifugation and flushed with methanol to remove the surface attached Pt(acac)$_2$. The resulted product was dried at 80 °C in a vacuum oven for 24 h, and then reduced in 5% H$_2$ at 150 °C for 1 h to obtain Pt$_1$-PMo@MIL-101.

**Synthesis of Pt$_1$-PMo@HKUST-1.** In all, 25 mg Pt(acac)$_2$, 0.25 g Cu(NO$_3$)$_2$·3H$_2$O, 0.3 g H$_3$PMo$_{12}$O$_{40}$·nH$_2$O, and 0.23 g H$_3$BTC were dissolved in 50 mL ethanol. The solution was continuously stirred for 12 h at room temperature. Precipitates appeared gradually. The precipitates were collected by centrifugation and washed through centrifugation and redispersion in alcohol and distilled water. After being dried at 80 °C in a vacuum oven for 24 h, the product was reduced to 5% H$_2$ at 150° C for 1 h to obtain Pt$_1$-PMo@HKUST-1.

**Synthesis of Pt$_1$-PMo@ZIF-67.** 0.75 g Co(NO$_3$)$_2$ • 6H$_2$O was dissolved in 25 mL methanol and 0.05 g H$_3$PMo$_{12}$O$_{40}$ • nH$_2$O was dissolved in 10 mL deionized water, then the two solutions were mixed and stirred for 30 min. A solution of 1.7 g 2-methylimidazole and 30 mg Pt(acac)$_2$ in 25 mL methanol was dumped into the abovementioned mixture during vigorous stirring. The resultant mixture was further stirred for another 4 h, and the precipitates were collected by centrifugation and washed through centrifugation and redispersion in methanol and distilled water. After being dried at 80 °C in a vacuum oven for 24 h, the product was reduced to 5% H$_2$ at 150 °C for 1 h to obtain Pt$_1$-PMo@ZIF-67.

**X-ray absorption spectra collection and data processing.** The X-ray absorption fine structure spectra data were collected at 1W1B station in Beijing Synchrotron Radiation Facility (BSRF, operated at 2.5 GeV with a maximum current of 250 mA. The XAFS data of Pt$_1$-PMo@MOFs samples and all the references (metal foils and oxide bulks) were recorded in transmission mode using an ionization chamber. All samples were pelletized as disks of 8 mm diameter.

The acquired EXAFS data were processed according to the standard procedures using the ATHENA module implemented in the IFEFFIT software packages. The EXAFS spectra were obtained by subtracting the post-edge background from the overall absorption and then normalizing with respect to the edge jump step. Then, $\chi(k)$ data in the $k$-space were Fourier transformed to real ($R$) space using hanging windows (d$k$ = 1.0 Å$^{-1}$) to separate the EXAFS contributions from different coordination shells. The quantitative information can be obtained by the least-squares curve fitting in the $R$ space with a Fourier transform $k$ space, using the module ARTEMIS of programs of IFEFFIT. The backscattering amplitude F($k$) and phase shift Φ($k$) were calculated using FEFF8.0 code.

**Catalytic diboration of phenylacetylene tests.** In a typical procedure, phenylacetylene (0.5 mmol), B$_2$pin$_2$ (0.5 mmol), and catalyst (20 mg) were placed in a Shlenck tube equipped with a stir bar, and then toluene (2.0 mL) was injected and the mixture was stirred at 100 °C for the corresponding reaction time. After the reaction was completed, the reaction mixture was analyzed by GC and GC-MS with dodecane as the internal standard. The overall TOF value was measured upon completion of reactions and the calculation of it was based on the total Pt loading in the catalyst.

**Liquid adsorption tests.** Phenylacetylene (0.1–0.5 mmol) or B$_2$pin$_2$ (0.1–0.5 mmol) was added to toluene (2 mL) to obtain the aimed substrate solutions with different concentrations. Then, MIL-101 (0.1 g) was added into the solutions respectively. After be continuously stirred for 1 h, the MIL-101 was separated and the substrate concentration was calculated from the results analyzed by gas

chromatography.

$$C_{\text{solution}} = M_{\text{before}} / V_{\text{solution}} \qquad (1)$$

$$C_{\text{MIL}-101} = (M_{\text{before}} - M_{\text{after}}) / V_{\text{MIL}-101} \qquad (2)$$

$C_{\text{solution}}$ is the substrate concentration in solution. $C_{\text{MIL-101}}$ is the substrate concentration in MIL-101 skeleton. $M_{\text{before}}$ is the mass of added substrate before adsorption. $M_{\text{after}}$ is the mass of substrate in the solution after adsorption which is calculated according to the GC results. $V_{\text{solution}}$ is the volume of the solution. $V_{\text{MIL-101}}$ is the space volume of MIL-101 which is 1.64 cm$^3$/g calculated from the cif file.

**Characterization**. TEM images were taken on a Hitachi HT7700 transmission electron microscope. The high-resolution TEM, HAADF-STEM images, and the corresponding energy-dispersive X-ray mapping were recorded by a JEOL JEM-2100F high-resolution transmission electron microscope operating at 200 kV. Aberration-corrected HAADF-STEM images are taken on a JEOL JEM-ARM200F TEM/STEM with a spherical aberration corrector working at 300 kV. ICP-OES was measured by Thermo Fisher IRIS Intrepid II. Nitrogen adsorption and desorption isotherms were obtained at 77 K using a Quantachrome Autosorb-1 instrument surface area analyzer. Powder X-ray diffraction patterns were measured with a Bruker D8 with Cu Kα radiation ($\lambda = 1.5406$ Å). XPS data were collected from a Thermo Fisher Scientific ESCALAB 250Xi XPS System, and the binding energy of the C1s peak at 284.8 eV was taken as an internal reference. FTIR spectroscopy was performed on a Bruker V70 infrared spectrometer in the frequency of 600–4000 cm$^{-1}$. The GC analysis was conducted on a Thermo Trace 1300 series GC with an FID detector using a capillary column (TR-5MS, from Thermo Scientific, length 30 m, i.d. 0.25 mm, film 0.25 μm). The GC-MS analysis was carried out on an ISQ GC-MS with an ECD detector (Thermo Trace GC Ultra) using a capillary column (TR-5MS, from Thermo Scientific, length 30 m, i.d. 0.25 mm, film 0.25 μm). H$_2$ temperature-programmed reduction was carried out on an AUTOCHEM 2920 chemisorption instrument.

## Data availability

The additional data are provided in the Supplementary Information. All the data that support the findings of this study are available from the corresponding author upon reasonable request.

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

## Acknowledgements

This work was supported by the National Natural Science Foundation of China (21901135, 21890383, 21871159, 11874036), the National Key R&D Program of China (2018YFA0702003), Science and Technology Key Project of Guangdong Province of China (2020B010188002), Basic Research Project of Shenzhen, China (JCYJ20200109142816479), and the Fundamental Research Funds for the Central Universities (DTU21LK18). We thank the BL14W1 station in Shanghai Synchrotron Radiation Facility (SSRF) and 1W1B station in BSRF for XAFS measurement.

## Author contributions

Yi.L., D.W., and Ya.L. conceived this work. Yi.L. performed the experiments, collected the data, and wrote the paper. X.W. and J.L. conducted the density functional theory calculation and analysis. Yi.L., Z.L., and J.Z. analyzed the data. Shou.L. and L.R.Z. helped with XANES and EXAFS spectrometry analyses. L.G. assisted in the AC HAADF-STEM characterization. Shu.L. helped with data analyses and discussions. All the authors contributed to the paper revisions. Yi.L. and X.W. contributed equally to this work.

## Competing interests

The authors declare no competing interests.
