## [Peer Review File · Nature Communications]

REVIEWER COMMENTS

Reviewer #1 (Remarks to the Author):

In the work by Liu and co-workers they proposed that they fabricated polyoxometalates (POMs) that stabilized atomically dispersed Pt sites inside different Metal Organic Frameworks (MOFs) including MIL-101, HKUST-1, and ZIF-67. The methodology proposed resulted interesting and creative; however, no enough evidences are presented and therefore I cannot recommend it for publication. See the comments below.

Line 60; conversional is conventional?

Regarding Figure 2 and the PXRD analysis. Can the authors describe the first peak at 4 degrees? What is this peak attributed to? Why there is a shift in the peaks when Pt is introduced into lower angles? The diffraction peaks appearing at around 5 degrees significantly varied upon PMo and PtPMo introduction but not explanation is given about it.

How can the authors be sure about PMo introduction? Maybe the MOF framework slightly changed due to the reaction and the N₂ isotherms, supplementary figure 2, are different because due the PMo could be covering the surface minimizing the surface of the MOF.

Additionally, if PMo would be successfully introduced, there should be a superlattice indication in the PXRD, are those peaks at 7 degrees corresponding to the superlattice? Can the authors provide the XRD of the PMo alone?

Could the authors provide a high quality figure 3a, the surface seems to exhibit rough surfaces (it can be just the data quality or data conversion). EDS maps are meaningless without the spectra as the Pt map seems very low in signal. Furthermore, images with more crystallites should be provide to obtain a more general idea of the product.

What is the error or the (ICP-OES), by observing the model proposed in figure 1 the Pt should not be higher than that?

Figure 3c is not clear enough to prove the presence of Pt, in fact I do not see any significant contrast variations within the marked circles.

The Pt-Pt peak of Pt foil standard at about 2.6 Å was not detected in Pt1-PMo@MIL-101, demonstrating the sole existence of isolated Pt single atoms.

This is the proof of that Pt foil is not present, Pt foil corresponds to a very high coordination number of Pt and the beautiful EXAFS data presented corroborates the small size of the Pt species but the existence of Pt isolated atoms (which could be present) are not demonstrated.

Regarding Figure S4. This figure and the correspondent EDS analysis are not sufficiently accurate to provide any relevant information. Image of the nps is not clear enough, as images are blurry and EDS lines should be provided together with the spectra. Furthermore, line profiles do not especially match with the figure. Line 167: Element line scan revealed that the surface of Pt NPs were protected by PMo; this statement is not supported by the experimental data.

When the PMo was absent, the authors show several MOF particles where clearly the metal is outside, a similar study should be presented for PtPMoMOF rather than one single image (in HAADF mode); However, no evidences of Pt nps or clusters inside the MOF are presented as they claim in line 169.

What is the experimental error of the XPS measurement? By looking at the second signal corresponding to PtPMoMIL-101 it seems there are two maxima next to each other, why do the authors picked the first one? (I am referring to the signal appearing at 236 eV.

Line 204, how was this Pt NPs@MIL-101 obtained . Was there the same amount of Pt in both materials?

Figure 4b circles instead of cycles.

Figure 4c. It is hard to see any of the intermediates proposed.

Line 214. The authors assume that the facets are {111} how do they know is they do not provide any evidence of this.

In figure S7, recovered corresponds to the first cycle or the fifth one. It is clear that the framework of the MOF has significantly decreased in terms of crystallinity.

Again there is not clear evidence of that the red circles in figure S8 correspond to Pt atoms.

How is the catalytic activity in comparison with other materials (in the literature) for the same reaction?

Reviewer #2 (Remarks to the Author):

This manuscript reports the preparation of single Pt atom anchored on polyoxometalate encapsulated inside various MOFs as catalyst for alkyne diborylation. The resulting encapsulated single atom Pt catalysts exhibit higher catalytic activity than related Pt catalysts. Controls revealed that the single Pt atom anchoring requires the encapsulation of polyoxometalate, since otherwise Pt nanoparticles are formed. The characterization of the material is convincing and catalytic data presented are in general sound. Publication in Nature Communications is recommended with some changes addressing the following comments:

- One of the main problems of single atom catalysis that also happens here is the low metal loadings. The reader would like to know what is the percentage of MIL-101 cage occupancy, polyoxometalate functionalization degree and what is the maximum Pt loading that can be achieved. 0.24% of Pt is not impressive and rather routine in the field. What is the PMo content?
- Related to the previous point, XPS Pt peak should be detected and analyzed. How can the authors claim to see the influence of Pt on the XPS Mo peak, when according to their models, only four out of 12 Mo atoms can be, in only a few of the total encapsulated POMs, have a Pt neighbor? If Pt is not directly

detectable, how can be discussed the minimal changes in the Mo peak? This is clearly a contradiction that needs to be solved.

- Fig 2c showing Pt dispersion should be improved and resolution enhanced to better show single Pt atoms.
- To support the authors' claim of reagent preconcentration within MIL-101 pores, adsorption data should be presented.
- Recyclability of the catalyst is improperly presented. Temporal profiles or initial reaction rates should be given as the parameter to prove catalytic stability. Otherwise, simply adding excessive catalyst amounts, coincident conversions at final time can be reached. In addition Fig. 4 b x-label "circles" should be corrected.
- Comparison between MIL-101 and zeolite Y should be cautious, since pore dimensions, pore volume and surface area of the two materials are different.
- When expanding their methodology to other MOFs, stability of HKUST-1 under the reaction conditions and recyclability should be presented. What is the pore window size of ZIF-67? How can PMo and reagents diffuse through the ZIF-67 internal pores?

If the authors address satisfactorily these issues, publication would be recommended.

Reviewer #3 (Remarks to the Author):

In this interesting work, the author prepare single atom Pt sites whose synthesis is favoured in the presence of polyoxometalates in the internal structures of different MOFs. Furthermore, they use the prepared catalysts in the phenylacetylene diboratin reaction, with promising results. The results, convincingly described, constitute a step forward in the synthesis and applications of single metal atom sites in heterogeneous catalysis. The manuscript provides enough information to support the claims and to allow for reproducibility. In my opinion, it can be published in Nature Commun. after some items are taken under consideration:

1. Line 109. It is "platinum-containing" instead of "platinum-contained".
2. Has it been checked that the reduction treatment at 150 °C is enough to reduce the platinum precursor to metallic platinum?
3. Line 180. Authors refer to a "Scheme 1", but it seems it is missing.
4. Line 177. Have the authors any proof that the PMOs can be reduced by hydrogen at only 150 °C?
5. Line 182. Which type of attraction could be acting between oxygen atoms in the reduced PMO and the Pt precursors, taking into account that these are neutral molecules (Pt(acac)₂)?
6. The authors conclude that PMOs are more easily reduced by hydrogen at 150 °C than the Pt precursor. Do you have any proof of this?
7. What about the stability of the Pt single atom sites?. In this study the catalytic reaction has been carried out at 100 °C. Could these catalysts be used in other reactions at higher temperatures?

Response to Comments

Manuscript Title: Fabricating Polyoxometalates-stabilized Single-Atom Site Catalysts in Confined Space with Enhanced Activity for Alkynes Diboration

Manuscript ID: NCOMMS-20-37254

Reviewer Comments:

Reviewer #1:

In the work by Liu and co-workers they proposed that they fabricated polyoxometalates (POMs) that stabilized atomically dispersed Pt sites inside different Metal Organic Frameworks (MOFs) including MIL-101, HKUST-1, and ZIF-67. The methodology proposed resulted interesting and creative; however, no enough evidences are presented and therefore I cannot recommend it for publication. See the comments below.

1. Line 60; conversional is conventional?

Response: We thank the reviewer for the suggestion on the wording. The typo has been corrected in the manuscript.

2. Regarding Figure 2 and the PXRD analysis. Can the authors describe the first peak at 4 degrees? What is this peak attributed to? Why there is a shift in the peaks when Pt is introduced into lower angles? The diffraction peaks appearing at around 5 degrees significantly varied upon PMo and PtPMo introduction but not explanation is given about it.

Response: We thank the reviewer for the inquiry of PXRD analysis. According to the crystal structure of MIL-101, we can simulate the PXRD pattern of MIL-101, and the peak at 3.98 degrees can be attributed to the facet of {400}. For a crystalline pores structure, when some guest molecules are introduced into the skeleton, the diffraction peaks will be affected, especially the guest molecules show an orderly arrangement in the pores. In fact, Ferey et al. have observed the changes in PXRD peak intensities after the incorporation of POMs in MIL-101 when they firstly reported this MOF (Science 2005, 309, 2040). Hatton et al. had carefully explored the arrangement of Keggin-type POMs in different pores of MIL-101 (Chem. Mater. 2012, 24, 1664). As they reported, when Keggin-type POMs are introduced into MIL-101, they are spontaneous and orderly arranged in the pores which can be reflected in the changes in PXRD. In our work, the variation of diffraction peaks is consistent with the literature. In fact, a similar phenomenon is also observed in other POM-based MOFs, such as Keggin-type POMs loaded in HKUST-1 (Science 1999, 283, 1148; J. Am. Chem. Soc. 2009, 131, 1883; ChemCatChem 2013, 5, 3086). We have cited relevant references in our manuscript. Thanks again to the reviewer for the careful reminder, which lead to a deeper understanding of the structural changes in the material.

3. How can the authors be sure about PMo introduction? Maybe the MOF framework slightly changed due to the reaction and the N₂ isotherms, supplementary figure 2, are different because due the PMo could be covering the surface minimizing the surface of the MOF. Additionally, if PMo would be successfully introduced, there should be a superlattice indication in the PXRD, are those peaks at 7 degrees corresponding to the superlattice? Can the authors provide the XRD of the PMo alone?

Response: We thank the reviewer for the thoughtful suggestion. MIL-101 is firstly reported by prof. G. Ferey in 2005 (Science 2005, 309, 2040). In that paper, they have successfully incorporated Keggin polyanions (K₇PW₁₁O₄₀) into the large cages of MIL-101. TGA, XRD, N₂ sorption, NMR, and IR techniques are used to confirm the presence of Keggin ions within the pores. And in that work, they have observed the changes in PXRD affected by the incorporation of POMs. In fact, MIL-101 is stable enough under the reaction conditions and N₂ sorption. We also reported the controllable arrangement of Keggin-type POMs in different pores of MIL-101 through different synthesis processes (J. Mater. Chem. A 2017, 5, 9611). The peaks at 7 degrees indeed can be attributed to the orderly arrangement of PMo in the pores of MIL-101. As requested by the reviewer, we here provide the XRD of PMo alone. We can see that the PMo are introduced inside of the MOF not just attached on the surface.

PXRD patterns of PMo@MIL-101, PMo, and MIL-101.

4. Could the authors provide a high quality figure 3a, the surface seems to exhibit rough surfaces (it can be just the data quality or data conversion). EDS maps are meaningless without the spectra as the Pt map seems very low in signal. Furthermore, images with more crystallites should be provide to obtain a more general idea of the product.

Response: Thanks for the reviewer's comment. In fact, the archetype of Figure 3 is clear. However, due to the file size limitation, when we upload our manuscript, the images in the files are compressed, resulting in reduced clarity. We attached a high-quality Figure 3 as a separate file. In Figure 3a, the surface of Pt₁-PMo@MIL-101 is smooth with no nanoparticles. EDS maps are used to preliminarily characterize the dispersion state of Pt. Although the Pt content is low and the signal intensity is not as strong as Cr and Mo, we can still recognize the homogeneous dispersion of Pt in the MOF. ICP

and XAFS further confirm the existence of Pt and its structure. As requested by the reviewer, we provide more images in the manuscript (Supplementary Figure 3).

5. What is the error or the (ICP-OES), by observing the model proposed in figure 1 the Pt should not be higher than that?

Response: We thank this reviewer for the inquiry about Pt content. The Pt content is determined by ICP. We carried out ICP test of Pt₁-PMo@MIL-101 for 5 times, and the measured Pt content is between 0.23 wt% and 0.25 wt% with very little fluctuation. Hence, we finally take the average of the loading amount as 0.24 wt%. When PMo occupies only the small pores of MIL-101, the loading amount of PMo is about 16 wt% which is consistent with the ICP results. According to the crystal structure of PMo, there are 6 anchoring sites on a Keggin-type polyanion theoretically. However, due to the space limitation, the number of introduced metal precursors is controlled to avoid the formation of nanoparticles. The ratio of moles of metal precursors to polyanions should not exceed 1:1. Hence, the maximum load of Pt should not exceed 1.68 wt%. We now reach the Pt content of 0.24 wt%, and we are trying to further optimize the loading amount of Pt in the follow-up work.

6. Figure 3c is not clear enough to prove the presence of Pt, in fact I do not see any significant contrast variations within the marked circles.

Response: Thanks for the reviewer's comment. As our response above, the figures would be compressed during the uploading process. Hence, for more clear presentation, we provide a high-quality Figure 3 as a separate file. However, comparing with the previously reported carbon-based single atomic sites materials, owing to the relatively lower contrast differences between Pt and Mo, although the difference in brightness is not particularly noticeable, the Pt atoms are still distinguishable when we zoom in on the image. In addition, the brighter Pt atoms locating on PMo but not MOF skeleton do also make our observations more difficult unfortunately. EXAFS data can further confirm the single-atom state of Pt to support our conclusion.

7. The Pt-Pt peak of Pt foil standard at about 2.6 Å was not detected in Pt₁-PMo@MIL-101, demonstrating the sole existence of isolated Pt single atoms. This is the proof of that Pt foil is not present, Pt foil corresponds to a very high coordination number of Pt and the beautiful EXAFS data presented corroborates the small size of the Pt species but the existence of Pt isolated atoms (which could be present) are not demonstrated.

Response: We thank the reviewer for the insightful comment. The appearance of Pt-Pt peak at about 2.6 Å is an evidence of the existence of the interaction between two adjacent Pt atoms, which is not just existed in Pt foil but also other Pt species including Pt clusters. Hence, the disappearance of the Pt-Pt peak is an important evidence to determine isolated single atomic sites are constructed. In addition, the existence of Pt-O peak and the relevant EXAFS fitting results both confirm the dispersion state of Pt as isolated single atoms. In fact, this evidence has been widely used for the

characterization of isolated single-atomic sites catalyst, for example single Pt atom on FeO_x (Nat. Chem. 2011, 3, 634); single Pt atom on CeO_2 (Science 2017, 358, 1419); and single Pt, Pd, Ru, Ir, Mo, Ga, Cu, Ni, Mn atoms on nitrogen-doped carbon (Nat Chem 2020, 12, 764) etc.

8. Regarding **Figure S4**. This figure and the correspondent EDS analysis are not sufficiently accurate to provide any relevant information. Image of the nps is not clear enough, as images are blurry and EDS lines should be provided together with the spectra. Furthermore, line profiles do not especially match with the figure. Line 167: Element line scan revealed that the surface of Pt NPs were protected by PMo; this statement is not supported by the experimental data.

Response: We thank the reviewer for this comment. Actually, using POMs as protecting ligands to synthesize metal nanoparticles have been reported, in which the structures of POM ligand shells on metal nanoparticles are widely accepted (Chem. Soc. Rev. 2012, 41, 7479). In our work, in order to demonstrate the function of MOFs during the fabrication of isolated single Pt sites, we synthesized PMo-protected Pt NPs. Despite a large amount of PMo were added as protecting ligands, we still cannot avoid the generation of Pt NPs which can be recognized as bright spots with the diameter between 2 and 3 nm, although the particle boundaries are not particularly distinct limited by the resolution of TEM equipment. Element line scan is used to verify the POM-protected structures that have been widely accepted. The two regions marked in the figure below are illustrative for the Pt core POM shell structure. As shown in the figure, the signal of Pt element is obviously higher in the center of the particle, indicating that the core of the particle is composed of Pt. However, when scanning to the boundary of the particle, the signal of Pt decreased accompanied by the signal of Mo increased obviously, which demonstrate the shell of the nano-particle constructed by POM.

Supplementary Figure 5. Element line scan of Pt NPs@ PMo.

9. When the PMo was absent, the authors show several MOF particles where clearly the metal is outside, a similar study should be presented for PtPMoMOF rather than one single image (in HAADF mode); However, no evidences of Pt nps or clusters inside the MOF are presented as they claim in line 169.

Response: We thank the reviewer for the insightful comment. The synthesis process of Pt NPs@MIL-101 are similar with Pt₁-PMo@MIL-101 just without the incorporation of PMo in the skeleton. During the synthesis process, the metal precursors are first adsorbed into the pores of MIL-101, then reduced by H₂. Owing to the high mobility of Pt atoms and the shortage of anchoring sites on MOFs skeleton, part of the Pt atoms would diffuse from the pores of the MOFs to the surface. This phenomenon is common in the fabrication of metal NPs in MOFs (Chem. Soc. Rev., 2017, 46, 4774). Hence, the nano clusters and NPs are existence both in the cages and on the outside surface of MIL-101. We added two images (Supplementary Figure 6) to give more clear evidence as the reviewer requested.

10. What is the experimental error of the XPS measurement? By looking at the second signal corresponding to PtPMoMIL-101 it seems there are two maxima next to each other, why do the authors picked the first one? (I am referring to the signal appearing at 236 eV).

Response: We thank the reviewer for this comment. The data are calibrated using C1s peaks. In fact, the existence of two maxima around 236 eV was caused by the bottom noise signal, and we did not pick either of them. Owing to the fixed interval (3.2 eV) between the two peaks of Mo, despite the impact of the noise we can determine the exact position according to the peak around 232.6 eV. Therefore, the peak position can be identified.

11. Line 204, how was this Pt NPs@MIL-101 obtained. Was there the same amount of Pt in both materials?

Response: Thanks for the reviewer's comment. The synthesis method of Pt NPs@MIL-101 is provided in the supplementary information. The synthesis process is similar with Pt₁-PMo@MIL-101 just without the incorporation of PMo in the skeleton. We also tested the Pt content in Pt NPs@MIL-101 by ICP. Owing to the same feeding ratio of Pt(acac)₂, the obtained Pt content of Pt NPs@MIL-101 is 0.24 wt% which is the same with Pt₁-PMo@MIL-101. Relevant descriptions have been added to the supplementary information.

12. Figure 4b circles instead of cycles. Figure 4c. It is hard to see any of the intermediates proposed.

Response: We thank the reviewer for these comments. "Circles" has been corrected to "Cycles" in Figure 4b. The compression of the picture during file upload process causes a decrease in resolution, so we increase the size of the intermediates for better display. Figure 4 has been updated in the manuscript.

Fig. 4 b The cyclic catalytic performance of Pt₁-PMo@MIL-101. **c** The calculated energy evolution of Pt₁-PMo@MIL-101 and Pt (111) catalyzed diboration reactions.

13. Line 214. The authors assume that the facets are {111} how do they know is they do not provide any evidence of this.

Response: We thank the reviewer for the query of the formation of Pt NPs. We carried out HRTEM to determine the facets of Pt NPs in MIL-101. As shown in supplementary figure 6, the lattice fringe spacing is 0.226 nm, which indicate that the facets of Pt NPs are {111}. The relative evidence has been added in supplementary information.

Supplementary Figure 6. TEM and HRTEM images of Pt NPs@MIL-101 without PMo.

14. In figure S7, recovered corresponds to the first cycle or the fifth one. It is clear that the framework of the MOF has significantly decreased in terms of crystallinity.

Response: We thank this reviewer for the inquiry about the recovered catalyst. The original Figure S7 has been renumbered as Supplementary Figure 9, in which the recovered catalyst was the one used after five cycles. We can see that the main diffraction peaks are still present, implying that the structure of the catalyst is not destroyed. The decreased intensity of the diffraction peaks might be caused by the incorporation of guest molecules in the pores or the change in catalyst size during the catalytic reaction process, but the structure of the catalyst is still maintained.

15. Again there is not clear evidence of that the red circles in figure S8 correspond to Pt atoms.

Response: Thanks for the reviewer's comment again. As our response above, the figures were compressed during the uploading process. The Figure S8 has been renumbered as Supplementary Figure 10. We also provide a high-quality Supplementary Figure 10 as a separate file, in which the isolated single Pt atoms are distinguishable when we zoom in on the image.

16. How is the catalytic activity in comparison with other materials (in the literature) for the same reaction?

Response: Thanks for the reviewer's comment. Up to now, the reported heterogeneous catalysts for such diboration reactions include nanoporous-gold, Pt/TiO₂, and Pt/MgO. All these catalysts exhibited low catalytic efficiency with TOF < 50 h⁻¹. The TOF of our catalyst reach up to 1017 h⁻¹. Relevant description and literatures have been added in the manuscript.

Reviewer #2 (Remarks to the Author):

This manuscript reports the preparation of single Pt atom anchored on polyoxometalate encapsulated inside various MOFs as catalyst for alkyne diborylation. The resulting encapsulated single atom Pt catalysts exhibit higher catalytic activity than related Pt catalysts. Controls revealed that the single Pt atom anchoring requires the encapsulation of polyoxometalate, since otherwise Pt nanoparticles are formed. The characterization of the material is convincing and catalytic data presented are in general sound. Publication in Nature Communications is recommended with some changes addressing the following comments:

1. One of the main problems of single atom catalysis that also happens here is the low metal loadings. The reader would like to know what is the percentage of MIL-101 cage occupancy, polyoxometalate functionalization degree and what is the maximum Pt loading that can be achieved. 0.24% of Pt is not impressive and rather routine in the field. What is the PMo content?

Response: We thank the reviewer for the thoughtful suggestion which we totally agree. Indeed, low metal loading is a common problem for single-atom catalysts. In our work, owing to the pore size limitation, PMo can only occupy the small pores (cage A in Figure 2a) of MIL-101, and the content of PMo is about 16 wt%. According to the crystal structure of PMo, there are 6 anchoring sites on a Keggin-type polyanion theoretically. However, due to the space limitation, the number of introduced metal precursors is controlled to avoid the formation of nanoparticles. The ratio of moles of metal precursors to polyanions should not be exceeded 1:1. Hence, the maximum load of Pt should not exceed 1.68 wt%. We now reach the Pt content of 0.24 wt% and we are trying to further optimize the loading amount of Pt in the follow-up works. Relevant descriptions have been added to the manuscript.

2. Related to the previous point, XPS Pt peak should be detected and analyzed. How can the authors claim to see the influence of Pt on the XPS Mo peak, when according to their models, only four out of 12 Mo atoms can be, in only a few of the total encapsulated POMs, have a Pt neighbor? If Pt is not directly detectable, how can be discussed the minimal changes in the Mo peak? This is clearly a contradiction that needs to be solved.

Response: We thank the reviewer for the discussion about XPS. In fact, we initially tried to detect the signal from Pt but failed, which may be due to the low Pt loading resulting in insufficient signal intensity and the fact that XPS technique can only detect the elements on the surface of the sample. In turn, we tried to detect the signal from Mo, because Yan et al. reported the influence of Pt located on PMo to the signal of Mo although the Pt content is only 0.9 wt% in their work (*Angew. Chem. Int. Ed.* 2016, 55, 8319). Fortunately, we did detect this subtle change. We also feel confused at first, but after deep thinking, we speculate that although the number of Mo atoms affected by Pt is relatively limited, the interaction is objective, and each Pt atom can affect up to four Mo atoms which make the signal and its shift affected by Pt more detectable like an amplifier. In fact, Yan et al. also collected Pt signal but with much worse quality. Owing to the lower Pt content in our material, we did not detect the signal of Pt unfortunately.

3. Fig 3c showing Pt dispersion should be improved and resolution enhanced to better show single Pt atoms.

Response: Thanks for the reviewer's comment. Due to the file size limitation, when we upload our manuscript, the images in the files are compressed, resulting in reduced clarity. Hence, for more clear presentation, we provide a high-quality Figure 3 as a separate file. Although owing to the relatively lower contrast differences between Pt and Mo, the brighter Pt atoms locating on PMo but not MOF skeleton do make our observations more difficult unfortunately, the Pt atoms are still distinguishable when we zoom in on the image. In addition, EXAFS data further confirm the single-atom state of Pt to support our conclusion.

4. To support the authors' claim of reagent preconcentration within MIL-101 pores, adsorption data should be presented.

Response: We thank the reviewer for the thoughtful suggestion. Because the liquid phase adsorption test cannot directly obtain the corresponding adsorption amount, the adsorption results are calculated by monitoring the concentration change of the solution. Specifically, phenylacetylene (0.1 to 0.5 mmol) or B₂pin₂ (0.1 to 0.5 mmol) was added to toluene (2 mL) to obtain the aimed substrate solutions with different concentration. Then, MIL-101 (0.1 g) were added into the solutions respectively. After be continuously stirred for 1 hour, the MIL-101 was separated and the substrate concentration was calculated from the results analyzed by gas chromatography. We can calculate the adsorbed amount of substrate according to the GC results. Combining the pore volumes of MIL-101 (1.64 cm³/g), we can obtain the substrate concentration inside MIL-101. Figure 4d and 4e is the calculated adsorption data. We here illustrate the testing and calculation process in detail with an example, when 0.5 mmol phenylacetylene is added in 2 ml toluene, the concentration is $102.14 \times 0.5 / 2 = 25.5$ mg/ml. After reaching adsorption equilibrium, we can monitor the concentration changing to 85.7% of the initial value by GC. According to this result, we can calculate the adsorbed phenylacetylene is 7.293 mg. And then the relevant substrate concentration in MIL-101 can be calculated as 44.47 mg/ml, which is obvious higher than that in the solution. The same calculation method is used for other concentration conditions, and the details have been added in the part of METHOD of the manuscript. Figure 4d and 4e are plotted based on the corresponding calculation results.

5. Recyclability of the catalyst is improperly presented. Temporal profiles or initial reaction rates should be given as the parameter to prove catalytic stability. Otherwise, simply adding excessive catalyst amounts, coincident conversions at final time can be reached. In addition Fig. 4 b x-label "circles" should be corrected.

Response: We thank the reviewer for the suggestion about the recyclability of the catalyst. As the reviewer request, we monitored the conversion in different cycles, and plotted the temporal profiles which showed no obvious change. This figure has been added in the Supplementary Information file as Supplementary Figure 11. In Fig. 4b, the x-label "circles" has been corrected to "cycles".

Supplementary Figure 11. The temporal catalytic performance of recovered Pt₁-PMo@MIL-101 in different cycles.

6. Comparison between MIL-101 and zeolite Y should be cautious, since pore dimensions, pore volume and surface area of the two materials are different.

Response: Thanks for the reviewer's constructive comment. As the reviewer mentioned, the differences in pore dimensions, pore volume and surface area between MIL-101 and zeolite Y all would influence their catalytic activities. However, although the existence of Pt species in zeolite Y, these catalysts exhibit no catalytic activity, and this essential difference prompts us to consider the role of the composition of the carrier in the reaction not just spatial structure. Hence, our aim is to make a qualitative analysis by this comparison without specifically comparing the performance of the two catalysts. Many thanks to the reviewer for such insightful advice.

7. When expanding their methodology to other MOFs, stability of HKUST-1 under the reaction conditions and recyclability should be presented. What is the pore window size of ZIF-67? How can PMo and reagents diffuse through the ZIF-67 internal pores?

Response: We thank this reviewer for the inquiry about the expansion of the synthetic methodology. HKUST-1 is stable enough under our reaction conditions, PXRD (Supplementary Figure 16) confirmed the structural integrity of the MOFs. In fact, the introduction of POMs can improve the stability of the framework which has been confirmed in our other works and the recyclability is well (J. Am. Chem. Soc. 2015, 137, 12697, Chem. Commun. 2014, 50, 10023, ChemCatChem 2013, 5, 3086). For ZIF-67, the pore window size is about 6Å which is smaller than Keggin-type POM. Hence POMs are incorporated by one-step synthesis method but not post-diffusion. The detailed synthesis method is described in the "METHODS" section of the manuscript.

Reviewer #3 (Remarks to the Author):

In this interesting work, the author prepare single atom Pt sites whose synthesis is favoured in the presence of polyoxometalates in the internal structures of different MOFs. Furthermore, they use the prepared catalysts in the phenylacetylene diborane reaction, with promising results. The results, convincingly described, constitute a step forward in the synthesis and applications of single metal atom sites in heterogeneous catalysis. The manuscript provides enough information to support the claims and to allow for reproducibility. In my opinion, it can be published in Nature Commun. after some items are taken under consideration:

1. Line 109. It is “platinum-containing” instead of “platinum-contained”.

Response: We thank the reviewer for the suggestion on the wording. We have corrected it in the manuscript.

2. Has it been checked that the reduction treatment at 150 °C is enough to reduce the platinum precursor to metallic platinum?

Response: Thanks for the reviewer’s comment. To determine the reduction temperature of Pt(acac)₂, we carried out H₂ temperature-programmed reduction, and the profiles are provided below. For MIL-101, no reduction peak is observed under 220 °C. After loading Pt(acac)₂, a reduction peak at approximately 150 °C appeared, indicating the reduction of Pt precursor. More interestingly, the reduction temperature decreased about 30 °C after the introduction of PMo, which was consistent with the reduction process we postulate. In fact, Yan et al. also observed similar phenomenon in their work (Angew. Chem. Int. Ed. 2016, 55, 8319). Relevant descriptions have been added in the manuscript.

Supplementary Figure 8. H₂-TPR profiles of Pt(acac)₂@MIL-101 and Pt(acac)₂-PMo@MIL-101.

3. Line 180. Authors refer to a “Scheme 1”, but it seems it is missing.

Response: Thanks for the reviewer’s constructive comment. It is a typo. It should be “Figure 1”, and we have corrected it in the manuscript.

4. Line 177. Have the authors any proof that the PMos can be reduced by hydrogen at only 150 °C?

Response: We thank this reviewer for the inquiry about the reduction of PMo. We reduced PMo at 150 °C for one hour, and tested the UV-Vis spectrum of the product. A distinct absorption peak around 800 nm was detected which can be attributed to the formation of heteropoly blues (PMo_{red}). This phenomenon demonstrated that PMo can be reduced by H₂ at 150 °C.

UV-Vis spectra of PMo and PMo reduced by H₂ at 150 °C for 1 hour.

5. Line 182. Which type of attraction could be acting between oxygen atoms in the reduced PMO and the Pt precursors, taking into account that these are neutral molecules Pt(acac)₂?

Response: We thank the reviewer for the comment. We speculate that during the reduction process, the acac⁻ ligands on Pt(acac)₂ would be removed, and then the Pt precursors would be attracted by polyanions through electrostatic interactions. Hence, PMO_{red} is more favorable than PMo because of its higher anion charge number. We think that this can also be interpreted as a ligand exchange process, since polyanions are available as inorganic ligands.

6. The authors conclude that PMOs are more easily reduced by hydrogen at 150 °C than the Pt precursor. Do you have any proof of this?

Response: Thanks for the reviewer's comment. As we mentioned above, the reducing temperature for Pt(acac)₂ decreased about 30 °C after the introduction of PMo. Hence we tried to reduce PMo at 120 °C, and we found that part of PMo was reduced to PMO_{red} which can be identified by the color change. In addition, the reduced PMo can be oxidized back to PMo by H₂O₂ which proved the successful reduction of PMo at 120 °C.

The pictures of PMo before and after reduced by H₂, and then the PMO_{red} was oxidized by H₂O₂.

7. What about the stability of the Pt single atom sites? In this study the catalytic reaction has been carried out at 100 °C. Could these catalysts be used in other reactions at higher temperatures?

Response: We thank this reviewer for the inquiry about the stability of the Pt single atom sites. The catalyst is stable at 100°C in this work, and we also trying to expand the application of this kind of catalyst. Recently, we have used this kind of catalysts in gas-solid reactions, and the single atom sites are stable under 160°C. Related results will be reported in the near future.

REVIEWER COMMENTS

Reviewer #1 (Remarks to the Author):

After the revision, significant improvements have been made in the manuscript answering some of the points addressed and therefore I would recommend it for publication after some additional corrections. Figure 6, authors describe as Pt{111} facets. They are actually measuring lattice fringes, so it would be more accurate to describe as a lattice fringe rather than a facet as the image is no clear enough.

Regarding EDS analyses, EDS spectra should be provided (as supplementary information) to see where the maps and line profiles have been extracted from.

I am still not fully convinced about the STEM-ADF information presented in figure 3 (and other figures). By looking at the atomic number of Mo (42), Cr (24) and Pt(78); the latest should be observed more clearly. Therefore, clearer STEM-ADF needs to be provided, considering that for this observation MOF framework is not an issue, more clear data is necessary and it can be achieved (there are many reports dealing with the visualization of single atoms Pt, Ir or Au for example on similar supports).

There is an additional aspect that I am not fully convince about and it is the significant difference between the XRD of the fresh and on the recovered material. By looking at figure 11, after five cycles the performance keeps stable. However, PXRD displays significant differences and the response given by the authors is not fully convincing. I suggest recording additional electron microscopy analyses by TEM; to evaluate the MOF morphology, to observe if there has been particle formation, or (by simple electron diffraction) evaluate the crystallinity of several particles. As there is some aspects that are affecting the PXRD results and do not affect the performance of the material.

Reviewer #2 (Remarks to the Author):

The authors have try to address my previous comments. Some of them, like recyclability have been properly solved. However, from the author's response, I still think that Pt loading is low and should be increased. One Pt every out of six possible anchoring sites is not impressive. No efforts to increase this low loading have been made. Incidentally, the authors still do not indicate what is the percentage of cages that are occupied by Pt.

I am still not convinced that a minor (undetectable) Pt loading of one every six single occupancy can result in a detectable influence on Mo. Just, because there was a previous report does not answer the question of how an indirect detection of Pt based on Mo can be possible when the vast majority of Mo atoms are not neighbors of Pt.

Regarding the use of ZIF-67, the authors consider that Pt1-PMo is incorporated inside the sodalite-like cages, but the point is that reagents cannot diffuse through the small cage windows with a diameter of 0.6 nm through which only small gases can access. Certainly. the reagents should not be able to diffuse through the pores. In addition, I think that stronger evidence of HKUST-1 stability should be provided. Therefore, I think that the current version still needs to be improved before acceptance.

Reviewer #3 (Remarks to the Author):

The authors have taken into account the comments by the reviewers, and the paper has been improved accordingly. In my opinion, it can now be published in Nature Communications in its present form.

Response to Comments

Manuscript Title: Fabricating Polyoxometalates-stabilized Single-Atom Site Catalysts in Confined Space with Enhanced Activity for Alkynes Diboration

Manuscript ID: NCOMMS-20-37254A

Reviewer Comments:

Reviewer #1:

After the revision, significant improvements have been made in the manuscript answering some of the points addressed and therefore I would recommend it for publication after some additional corrections.

1. Supplementary Figure 6, authors describe as Pt {111} facets. They are actually measuring lattice fringes, so it would be more accurate to describe as a lattice fringe rather than a facet as the image is no clear enough.

Response: We thank the reviewer for the thoughtful suggestion. We have corrected the description in Supplementary Figure 6 by replacing crystal plane with lattice fringe to make the expression more accurate. The corresponding modified parts have been marked in supplementary information.

2. Regarding EDS analyses, EDS spectra should be provided (as supplementary information) to see where the maps and line profiles have been extracted from.

Response: We thank this reviewer for the inquiry about the EDS spectra. As requested by the reviewer, we add EDS spectra of Pt₁-PMo@MIL-101 and Pt NPs@PMo in Supplementary Figure 3 and Supplementary Figure 5 respectively. The corresponding spectra are also attached below.

Supplementary Figure 3. EDS spectra of Pt₁-PMo@MIL-101 corresponding to Figure 3b.

Supplementary Figure 5. EDS spectra of Pt NPs@PMo.

3. I am still not fully convinced about the STEM-ADF information presented in figure 3 (and other figures). By looking at the atomic number of Mo (42), Cr (24) and Pt (78); the latest should be observed more clearly. Therefore, clearer STEM-ADF needs to be provided, considering that for this observation MOF framework is not an issue, more clear data is necessary and it can be achieved (there are many reports dealing with the visualization of single atoms Pt, Ir or Au for example on similar supports).

Response: Thanks for the reviewer's comment. As requested by the reviewer, we devoted a lot of effort to repeatedly testing the aberration-corrected HAADF STEM images of Pt₁-PMo@MIL-101 more carefully, and the resolution of Pt single atoms has indeed been improved. The related image Figure 3c has been replaced by a clearer image and also attached below. We also compared our new image with the reported Pt₁ on Mo-based supports in the literature (Figure R1), the contrast is comparable now. In addition, we carefully analyze the location of Pt single atoms on the surface of PMo sphere, and found that due to the Td symmetry of Keggin-type PMo, regardless of the direction from which we observe, compared to the projection of single Pt atom, the projection of surrounding Mo atoms is always overlapped with two or more atoms, which make the observation of single Pt atom more difficult than other Mo-based supports.

Figure 3c. Aberration-corrected HAADF STEM image of Pt₁-PMo@MIL-101.

Figure R1. Aberration-corrected HAADF STEM images of Pt₁/α-MoC in references.

Figure R2. The structure of Pt₁ on the surface of PMo viewing from different direction.

4. There is an additional aspect that I am not fully convince about and it is the significant difference between the XRD of the fresh and on the recovered material. By looking at figure 11, after five cycles the performance keeps stable. However, PXRD displays significant differences and the response given by the authors is not fully convincing. I suggest recording additional electron microscopy analyses by TEM; to evaluate the MOF morphology, to observe if there has been particle formation, or (by simple electron diffraction) evaluate the crystallinity of several particles. As there are some aspects that are affecting the PXRD results and do not affect the performance of the material.

Response: Thanks to the reviewer's suggestion. As requested by the reviewer, we further characterized the recovered Pt₁-PMo@MIL-101 catalyst by SEM, TEM and STEM (Supplementary Figure 10). From these results, we can find that no Pt nanoparticles were formed inside the catalyst after the catalytic reaction, but we also observed that a small portion of the catalyst was broken from SEM image. This breakage of recovered catalyst could be explained by the mechanical stirring. Based on these results, we can deduce that the change in PXRD is not due to Pt agglomeration inside the MOF skeleton. We speculate that this change may be caused by the residual of products or solvent molecules inside the pores of the MOFs, which does not affect the reproduction of the catalytic performance. The additional characterizations have been added into Supplementary Figure 10 which also attached blow.

Supplementary Figure 10. SEM, TEM, HAADF-STEM, and Aberration-corrected HAADF-STEM images of recovered Pt₁-PMo@MIL-101.

Reviewer #2:

The authors have try to address my previous comments. Some of them, like recyclability have been properly solved.

1. However, from the author's response, I still think that Pt loading is low and should be increased. One Pt every out of six possible anchoring sites is not impressive. No efforts to increase this low loading have been made. Incidentally, the authors still do not indicate what is the percentage of cages that are occupied by Pt.

Response: We thank the reviewer for the thoughtful suggestion. As the reviewer's request, we tried our best to increase the Pt single atom loading. By increasing the Pt precursor feeding, the Pt loading was improved to a certain extent, and finally the Pt loading reached about 0.6 wt% in Pt₁-PMo@MIL-101. When we further introduce more Pt precursors to about 0.8 wt%, Pt nanoparticles are observed during the reduction process (Figure R3).

Figure R3. HAADF-STEM images of 0.4 wt% Pt₁-PMo@MIL-101, 0.6 wt% Pt₁-PMo@MIL-101, and 0.8 wt% Pt NPs-PMo@MIL-101, respectively.

We carried out N₂ adsorption/desorption isotherms of Pt₁-PMo@MIL-101. Comparing with PMo@MIL-101, the loading of Pt single atoms does not significantly affect the porosity of the material (Supplementary Figure 2). Obviously, the reduction of the porosity of MIL-101 skeleton is mainly due to the introduction of PMo, about half of the pore space of MIL-101 are occupied by

PMo. But according to the loading amount of Pt single atoms and the location sites of Pt₁ on PMo, we can calculate that Pt single atoms are present in about 10% pores of MIL-101 skeleton. The related N₂ adsorption/desorption results are added in Supplementary Figure 2 and also attached below.

Supplementary Figure 2. N₂ adsorption/desorption isotherms of MIL-101, PMo@MIL-101, and Pt₁-PMo@MIL-101 respectively.

2. I am still not convinced that a minor (undetectable) Pt loading of one every six single occupancy can result in a detectable influence on Mo. Just, because there was a previous report does not answer the question of how an indirect detection of Pt based on Mo can be possible when the vast majority of Mo atoms are not neighbors of Pt.

Response: We thank the reviewer for the discussion about XPS. Probably due to the detection limitation of the instrument we used before, we could not detect the signal of Pt, so we tried to collect the signal of Pt on a different instrument this time. After some efforts, we finally got the XPS data of Pt 4f for Pt₁-PMo@MIL-101 and Pt NPs@MIL-101, respectively (Supplementary Figure 7). By comparison, we can see that the binding energy of Pt in Pt₁-PMo@MIL-101 is obviously higher than that in Pt NPs@MIL-101, which indicates that the coordination environment of Pt in Pt₁-PMo@MIL-101 is different from that in Pt NPs@MIL-101. And the results also confirm that Pt₁ has a positive charge, which is consistent with the XANES data (Figure 3d). However, this result can not directly prove the interaction between Pt single atoms and PMo. So we performed a detailed analysis of the XPS results for Mo 3d. Although each Pt₁ is coordinated to only 4 O atoms on the surface of PMo, but considering the surface charge of Keggin-type polyoxometalates is off-domain due to their spherical structure, so the introduction of a single Pt atom on the surface of PMo would affect the whole electrons distribution of the Keggin-type polyoxometalate, in other words all 12 Mo atoms within a PMo would be affected even part of them are not surround the Pt₁ atom. Taking into account that the proportion of affected PMo can reach close to 20% with one Pt atom anchored on the surface of each PMo, it is reasonable that small signal changes can be detected in Mo XPS spectra. The relevant description is added to the revised manuscript.

Supplementary Figure 7. Pt 4f XPS spectra of Pt NPs@MIL-101 and Pt₁-PMo@MIL-101.

3. Regarding the use of ZIF-67, the authors consider that Pt₁-PMo is incorporated inside the sodalite-like cages, but the point is that reagents cannot diffuse through the small cage windows with a diameter of 0.6 nm through which only small gases can access. Certainly, the reagents should not be able to diffuse through the pores. In addition, I think that stronger evidence of HKUST-1 stability should be provided.

Response: We thank this reviewer for the inquiry about the synthetic methodology. In fact, the specific synthesis steps of Pt₁-PMo@ZIF-67 are different from Pt₁-PMo@MIL-101. We have taken into account about the smaller apertures of ZIF-67 than MIL-101, therefore the introduction of both metal precursors and PMo is achieved by in-situ synthesis. Specifically, Co(NO₃)₂ • 6H₂O was dissolved in methanol and PMo was dissolved in deionized water, then the two solution were mixed and stirred for 30 min. A solution of 2-methylimidazole and Pt(acac)₂ in methanol was dumped into the abovementioned mixture during vigorous stirring. The resultant mixture was further stirred for another 4 hours, and the precipitates were collected by centrifugation and washed through centrifugation. PMo and Pt(acac)₂ are in-situ incorporated into the pores of ZIF-67 during the coordination process of Co²⁺ and 2-methylimidazole, which overcome the limitation of the aperture. The detailed synthesis method is described in the “METHODS” section of the manuscript.

Regarding the stability of HKUST-1, as requested by the reviewer, we tested the PXRD of different samples, including HKUST-1, PMo@HKUST-1, Pt(acac)₂-PMo@HKUST-1, and Pt₁-PMo@HKUST-1, at each step of the whole synthesis process (Figure R4). And these results demonstrated that the skeletal structure of these samples is always maintained. We also characterized different samples by SEM. After each reaction step (Figure R5), the morphology of the samples did not change significantly, indicating that the structure of the material was not destroyed and was stable. In fact, in as early as 2009, we have successfully incorporated different Keggin-type polyoxometalates into HKUST-1 and resolved their single crystal structures (denoted as NENU-n, J. Am. Chem. Soc. 2009, 131, 1883-1888). In which, the main framework composed of Cu²⁺ and BTC ligands is isomorphic to HKUST-1, and the Keggin-type polyoxometalates are selectively loaded in the pore A of HKUST-1 due to the symmetry match (Figure R6). As shown in Figure R4, we can see that the diffraction peak positions of the PMo-loaded HKUST-1 material are consistent with the simulated PXRD spectra, indicating that the structure of the main framework in the material is maintained, while the Keggin-type PMo is orderly loaded inside the pores of HKUST-1. When we introduced Pt precursors during the synthesis process, the diffraction peak position and

intensity of Pt(acac)₂-PMo@HKUST-1 did not change, indicating that the introduction of Pt precursors did not affect the crystallization of PMo@HKUST-1 and the structure of the main framework was maintained. After further reduction treatment, the diffraction peaks also maintained, indicating HKUST-1 was stable under the reaction conditions. The well-maintained morphology from HKUST-1 to Pt₁-PMo@HKUST-1 further prove the stability of these materials (Figure R5).

Figure R4. PXRD of HKUST-1, Simulated PMo@HKUST-1, PMo@HKUST-1, Pt(acac)₂-PMo@HKUST-1, and Pt₁-PMo@HKUST-1 respectively.

Figure R5. SEM images of (a) HKUST-1, (b) PMo@HKUST-1, (c) Pt(acac)₂-PMo@HKUST-1, and (d) Pt₁-PMo@HKUST-1 respectively.

Figure R6. The single crystal structure of PMo@HKUST-1.

Reviewer #3:

The authors have taken into account the comments by the reviewers, and the paper has been improved accordingly. In my opinion, it can now be published in Nature Communications in its present form.

Response: We thank this reviewer for the help of the improvement of our manuscript.

REVIEWER COMMENTS

Reviewer #1 (Remarks to the Author):

Comments to the authors:

Regarding Supplementary Fig 6. How the authors can claim (111) direction? Do they mean $d_{\{111\}}$ planes? and therefore 111 would correspond to Miller index? From the data presented, I do not know how they can claim particle orientation.

Despite the new electron microscopy data, combined with EXAFS and XANES, which was previously presented. The claiming are not supported by the experimental results. The authors compared their data with several nature papers of Pt supported on crystalline MoC supports. That does not imply that what the authors have circled are Pt atoms. Even though, the Pt signals from R1 are clearer than Figure 3, even in less favorable conditions as are crystalline supports which retrieve stronger signals than amorphous ones, as it is the current case. Could the authors plot an intensity profile to highlight the Pt signal respect to the Mo clusters?

In fact, by observing supplementary figure 10, there are agglomerates (yellow circle) that shows stronger signals than those identified as Pt atoms. I guess they correspond to those Mo clusters

Regarding the XPS Mo spectra, supplementary Fig 7, the way that authors identified the maxima of Pt-PMo@MIL-101 is ambiguous. The maxima of the first signal (lower energy) may be correct, but I do not understand how they chose the second signal as, between the two tinny maxima the one at higher energy is at least of equal intensity as the one selected by the authors to center the second peak, and it would be similar value as for PtMo@MIL-101.

There is another aspect regarding supplementary Fig.6 and DFT calculations. If the d-spacing corresponds to the {111} planes, that implies that what the authors are observing are not (111) facets. However, DFT calculations have been performed assuming Pt(111) facets. Would it be the same result on (110) or (100) facets?

I believe that all these issues should be addressed before publication. The authors have successfully obtained Pt-PMo@MIL-101 which show a remarkable activity, selectivity and stability. Surprisingly, Pt zeolites exhibited no conversion at all, which suggest the strong influence of the PMo or of the MOF or both together as when they studied PMo@MIL-101 did not show any catalytic activity. However, before publication in a high impact journal as Nature Communications discussion should be more precise.

Reviewer #2 (Remarks to the Author):

The authors have taken my previous comments in a constructive way and have provided additional evidences in support to their claims and conclusions. Particularly, the direct XPS detection of Pt with

observation of a notable shift in its binding energy respect to Pt nanoparticles is very convincing. Also they have clarified how Pt₁Mo can be included in ZIF-67 and provided evidence of HKUST-1 stability that is always a big concern. One of the limitations of single atom catalysis is the low loading and this issue is also present here with such a small Pt loading. However, I think that the manuscript is ready for publication in its present form.

Response to Comments

Manuscript Title: Fabricating Polyoxometalates-stabilized Single-Atom Site Catalysts in Confined Space with Enhanced Activity for Alkynes Diboration

Manuscript ID: NCOMMS-20-37254B

Reviewer #1 (Remarks to the Author):

Comments to the authors:

1. Regarding Supplementary Fig 6. How the authors can claim (111) direction? Do they mean $d\{111\}$ planes? and therefore 111 would correspond to Miller index? From the data presented, I do not know how they can claim particle orientation.

There is another aspect regarding supplementary Fig.6 and DFT calculations. If the d-spacing corresponds to the $\{111\}$ planes, that implies that what the authors are observing are not (111) facets. However, DFT calculations have been performed assuming Pt (111) facets. Would it be the same result on (110) or (100) facets?

Response: We thank the reviewer very much for the thoughtful suggestions. In order to further clarify the surface exposure of Pt NPs in MIL-101, we performed more detailed high-resolution TEM characterization of Pt NPs@MIL-101. Due to the relatively poor resolution of Pt NPs inside the MIL-101 framework, we chose Pt NPs that are on the surface of MIL-101 for detailed analysis. Here we added a new TEM image that shown in Supplementary Fig. 6, two sets of lattice fringes corresponding to the $\{111\}$ planes and one set of lattice fringes corresponding to the $\{100\}$ plane are observed. Combined with the surface morphology of this Pt nanoparticle, we can infer that for this nanoparticle, the six capped edges of the two-dimensional projected image of the nanoparticle are terminated by the (111) and (100) facets, where (111) facets are much more than (100) facets. However, in fact, we can observe from TEM images that for most of the Pt NPs in MIL-101, there is no clear morphology. Considering that Pt (111) facets has the lowest surface energy among the three common low index crystalline planes $\{111\}$, $\{100\}$ and $\{110\}$ of Pt NPs, therefore, if the growth of Pt nanoparticles is not artificially affected during the synthesis process, like this work, the surface of the obtained Pt nanoparticles should be predominantly composed of $\{111\}$ crystalline planes, which is a relatively common phenomenon in the growth of Pt nanoparticles. Hence, Pt (111) facets is chosen for DFT calculations. We also corrected the description in Supplementary Figure 6 to make the expression more accurate.

Supplementary Fig. 6 The measured lattice fringe spacing of Pt NPs@MIL-101

2. Despite the new electron microscopy data, combined with EXAFS and XANES, which was previously presented. The claiming are not supported by the experimental results. The authors compared their data with several nature papers of Pt supported on crystalline MoC supports. That does not imply that what the authors have circled are Pt atoms. Even though, the Pt signals from R1 are clearer than Figure 3, even in less favorable conditions as are crystalline supports which retrieve stronger signals than amorphous ones, as it is the current case. Could the authors plot an intensity profile to highlight the Pt signal respect to the Mo clusters?

In fact, by observing supplementary figure 10, there are agglomerates (yellow circle) that shows stronger signals than those identified as Pt atoms. I guess they correspond to those Mo clusters

Response: We thank this reviewer for the inquiry about the intensity profile of Pt atoms. As requested by the reviewer, we added Z-contrast analysis of Pt₁-PMo@MIL-101 in Figure 3c, and the enhanced contrast can be clearly observed which can be identified as single Pt atom. In addition, we also retested the aberration-corrected HAADF STEM images of recovered Pt₁-PMo@MIL-101 and provided corresponding Z-contrast analysis, in which single Pt atom can be identified (a new Supplementary Fig. 10). As marked by the reviewers, we indeed observed some aggregates, whose contrast were obviously lower than single Pt atom with the size around 1 nm, and we totally agree with the reviewer that these agglomerates correspond to Keggin-type PMo clusters.

Figure 3c. Aberration-corrected HAADF STEM images and corresponding Z-contrast analysis of Pt₁-PMo@MIL-101.

Supplementary Fig. 10 Aberration-corrected HAADF STEM images and corresponding Z-contrast

analysis of recovered Pt₁-PMo@MIL-101.

3. Regarding the XPS Mo spectra, supplementary Fig 7, the way that authors identified the maxima of Pt-PMo@MIL-101 is ambiguous. The maxima of the first signal (lower energy) may be correct, but I do not understand how they chose the second signal as, between the two tiny maxima the one at higher energy is at least of equal intensity as the one selected by the authors to center the second peak, and it would be similar value as for PtMo@MIL-101.

Response: We thank the reviewer for this comment. This is a very thoughtful question. In fact, the existence of two maxima around 236 eV was caused by the bottom noise signal, and we did not pick either of them. Owing to the fixed interval (3.2 eV) between the two peaks of Mo, hence once we determined the exact location of the first signal (lower energy), we can also determine the exact position of the second one only by adding 3.2 eV, even under the influence of the noise. Therefore, the peak position can be identified.

4. I believe that all these issues should be addressed before publication. The authors have successfully obtained Pt-PMo@MIL-101 which show a remarkable activity, selectivity and stability. Surprisingly, Pt zeolites exhibited no conversion at all, which suggest the strong influence of the PMo or of the MOF or both together as when they studied PMo@MIL-101 did not show any catalytic activity. However, before publication in a high impact journal as Nature Communications discussion should be more precise.

Response: We thank the reviewer for this constructive comment. Different supporters exhibited significant impact on the catalytic performance and we indeed discuss it in our manuscript. Considering that the Pt catalytic activity center was located inside the porous skeleton, and the reaction substrates need to enter the skeleton to contact with Pt sites, we investigated the adsorption abilities of MOF skeletons for phenylacetylene and B₂pin₂ under reaction conditions, and the concentrations of reactants on these selected substrates was all below the initial reaction concentrations to simulate the changes during the reaction process. As shown in Fig. 4d and 4e, MIL-101 exhibited conspicuous pre-concentration ability for both reactants, and this process was monitored by gas chromatography. After adsorption equilibrium reached, the calculated concentrations of reactants inside MIL-101 were higher than that in the residuary solution, even when the concentration decreased to only one-fifth of the initial. Obviously, in the inner space of the 3D skeleton, phenylacetylene and B₂pin₂ were “forced” to interact with single Pt atom sites, as they were confined in close proximity. In contrast, the pure inorganic support Y zeolite exhibited negligible adsorption for these two reactants, which should be responsible for the inert catalysis activities of Pt NPs@Y and Pt₁@Y. It is possible to attribute the signally disparate adsorption behaviors of MIL-101 and Y zeolite to their different chemical compositions. To be more specific, the organic component (terephthalic acid) of MIL-101 frameworks afforded hydrophobic-hydrophobic and aromatic-aromatic interactions with the organic reactants, which did not exist in Y zeolite. The relative discussion is included in the part of “Catalytic performance evaluation of Pt₁-PMo@MIL-101”.

Reviewer #2 (Remarks to the Author):

The authors have taken my previous comments in a constructive way and have provided additional evidences in support to their claims and conclusions. Particularly, the direct XPS detection of Pt with observation of a notable shift in its binding energy respect to Pt nanoparticles is very convincing. Also they have clarified how Pt1Mo can be included in ZIF-67 and provided evidence of HKUST-1 stability that is always a bid concern. One of the limitations of single atom catalysis is the low loading and this issue is also present here with such a small Pot loading. However, I think that the manuscript is ready for publication in its present form.

Response: We thank this reviewer for the help of the improvement of our manuscript.

REVIEWERS' COMMENTS

Reviewer #1 (Remarks to the Author):

The authors have addressed all the concerning issues and although there are still several aspects that would be subject of further investigation, the manuscript is more clear and precise now and therefore I recommend it for publication.